# Investigation of Radiation Effect on Structural and Optical Properties of GaAs under High-Energy Electron Irradiation

**DOI:** 10.3390/ma15175897

**Published:** 2022-08-26

**Authors:** Authit Phakkhawan, Aparporn Sakulkalavek, Siritorn Buranurak, Pawinee Klangtakai, Karnwalee Pangza, Nongnuch Jangsawang, Sawinee Nasompag, Mati Horprathum, Suphakan Kijamnajsuk, Sakuntam Sanorpim

**Affiliations:** 1Department of Physics, Faculty of Science, King Mongkut’s Institute of Technology Ladkrabang, Bangkok 10520, Thailand; 2Department of Physics, Faculty of Science, Khon Kaen University, Khon Kaen 40002, Thailand; 3Institute of Nanomaterials Research and Innovation for Energy (IN-RIE), Khon Kaen University, Khon Kaen 40002, Thailand; 4Thailand Center of Excellence in Physics, Chiang Mai University, P.O. Box 70, Chiang Mai 50202, Thailand; 5Gems Irradiation Center, Thailand Institute of Nuclear Technology, Nakhon-Nayok 26120, Thailand; 6Research Instrument Center, Khon Kaen University, Khon Kaen 40002, Thailand; 7National Electronics and Computer Technology Center (NECTEC), National Science and Technology Development Agency (NSTDA), Khlong Luang 12120, Thailand; 8National Metal and Materials Technology Center (MTEC), National Science and Technology Development Agency (NSTDA), Khlong Luang 12120, Thailand; 9Department of Physics, Faculty of Science, Chulalongkorn University, Phayathai Rd., Patumwan, Bangkok 10330, Thailand

**Keywords:** GaAs, electron irradiation, high-energy particle, hydrophilic

## Abstract

A systematic investigation of the changes in structural and optical properties of a semi-insulating GaAs (001) wafer under high-energy electron irradiation is presented in this study. GaAs wafers were exposed to high-energy electron beams under different energies of 10, 15, and 20 MeV for absorbed doses ranging from 0–2.0 MGy. The study showed high-energy electron bombardments caused roughening on the surface of the irradiated GaAs samples. At the maximum delivered energy of 20 MeV electrons, the observed root mean square (RMS) roughness increased from 5.993 (0.0 MGy) to 14.944 nm (2.0 MGy). The increased RMS roughness with radiation doses was consistent with an increased hole size of incident electrons on the GaAs surface from 0.015 (0.5 MGy) to 0.066 nm (2.0 MGy) at 20 MeV electrons. Interestingly, roughness on the surface of irradiated GaAs samples affected an increase in material wettability. The study also observed the changes in bandgap energy of GaAs samples after irradiation with 10, 15, and 20 MeV electrons. The band gap energy was found in the 1.364 to 1.397 eV range, and the observed intense UV-VIS spectra were higher than in non-irradiated samples. The results revealed an increase of light absorption in irradiated GaAs samples to be higher than in original-based samples.

## 1. Introduction

GaAs is a type of III-V semiconducting material composed of Ga element from column III and As element from column V of the periodic table of elements. This semiconducting material also includes InP, InAs, GaN, and InSb. GaAs is cubic crystals with a zincblende structure. It is the most studied and technologically utilized compound semiconductor material due to its several unique properties, such as its wider direct bandgap energy (1.42 eV at room temperature [1,2]), low exciton binding energy (4.2 meV) [3], and higher electron mobility (8800 cm^2^ V^−1^ s^−1^) [4]) compared to crystalline silicon. GaAs also exhibits light emitting [5], electromagnetic [6], and photovoltaic [7] properties. It can be utilized in high-speed semiconductor devices [8], high-power microwave and millimeter-wave devices [9], optoelectronic devices [5,10,11,12], medical detectors [13], and imaging devices [13,14,15,16]. Moreover, its bandgap energy can be tuned to the range appropriate for several applications, such as long wavelength emitters [17], detectors [18], and spintronic-related devices [17], by alloying it with other elements such as In, Al, Sb, and N to form InGaAs, AlGaAs, GaAsSb, and GaAsN, respectively, etc. These ternary alloys, InGaAs, AlGaAs, GaAsSb, and GaAsN, have been successfully grown by metal-organic chemical vapor deposition (MOCVD) [19,20,21,22] and by molecular beam epitaxy (MBE) [23,24,25,26]. Among its diverse applications, GaAs-based material can be used as a substrate for the epitaxial growth technology of other III-V semiconductors, including InGaAs, InGaN, GaAsN, and others. Moreover, one of the compound’s most essential applications is its use as the light-absorbing layer in high-efficiency multi-junction solar cells, which have great potential for space applications [27,28,29,30]. GaAs-based solar cells also have many advantages, such as high photoelectric conversion efficiency, good radiation resistance, and good performance at high temperatures [27,28,29,30]. Moreover, GaAs-based radiation detectors exhibit fast signal collections with sharp signal responses, high dynamic dose responses [13], and low electronic noise at room temperature in contrast to silicon [15,16]. GaAs also provides radiation tolerance and a cost-efficient that is promising to be applied for a photon-counting detector for medical X-ray imaging [15,16,31,32], fiber optic temperature sensor [33], and solid-state detector [34,35].

Using GaAs-based solar cells, GaAs-based radiation detectors, or other GaAs-based electronic devices in a radiation environment with high-energy radiation particles such as neutrons, protons, electrons, and others, must consider the harsh environment they are subjected to. Xu et al. [36] studied the effect of alpha-particle irradiation on the optical and electrical properties of InGaP/GaAs/Ge triple-junction solar cells for 5.1 MeV alpha particles with different fluences. They found that the photoluminescence intensity of the InGaP top cells and GaAs middle cells decreased with an increase in alpha-particle fluence. Shen and co-workers [37] studied Si-doped GaAs’ structural and photoelectric characteristics under different gamma irradiation doses (0, 0.1, 1, and 10 KGy). They found that the roughness of Si-doped GaAs film increases with the increasing irradiation dose, while the sample still has a good crystallinity. It was found that the luminescence intensity of Si-doped GaAs film was improved by reducing the non-radiation composite center of the GaAs layer after gamma irradiation. However, the generated current of Si-doped GaAs was reduced due to the decrease in carrier concentration and mobility. Gruginskie at el. [38] studied the effects of electron irradiation on the performance of GaAs solar cells under 1-MeV electrons with fluence up to 1 × 10^15^ e^−^ cm^−2^ in both experimental and theoretical analysis. They found that the incidence of electrons introduces lattice defects in the cells that act as recombination centers, directly impacting carrier lifetimes. A reviewed report found that radiation damage in solar cells causes a reduction in minority carrier diffusion length and thereby reduces overall conversion efficiency [39,40,41,42,43]. Based on the above reviews, irradiation by high-energy radiation particles degrade the physical properties of irradiated semiconductor materials. The irradiation of GaAs-based material by high-energy radiation particles can produce lattice defects in the form of vacancies, defect clusters, and dislocations [44,45,46,47,48]. The main effect of these lattice defects in GaAs-based material is the creation of recombination centers known to reduce the carrier minority lifetime and increase dark current. These can reduce the performance of GaAs-based devices in highly radiative environments, such as reduction of solar cell efficiency, detector sensitivity, and resolution. Thus, the nature of structural defects in GaAs-based material and its alloy in a high-energy particle environment must be well understood in microscopic and macroscopic views to know for engineering material preparation and optimize the sample structure for practical use. In our previous work [49], we studied the effect of gamma-ray irradiation on the structural property of GaAs_1−*x*_N*_x_* (N content (*x*) = 1.9 and 5.1% with various irradiation doses of 0, 0.5, 1.0, 1.5, and 2.0 MGy. After gamma-ray irradiation, we found that roughened surface with numerous holes and a cross-hatch pattern were observed on the GaAs_1−*x*_N*_x_* surface. The increase in N incorporation in both low- and high-N-content films after gamma-ray irradiation was attributed to the diffusion of N atoms at interstitial sites to either As lattice sites or vacancy sites during the irradiation process. The structural change in the irradiated GaAsN films came from atomic displacement caused by gamma-ray heating. To continue exploring the effect of a high-energy particle on the structural property of the GaAs-based material, we addressed the impact of high-energy electrons on GaAs’ structural properties in this work. Moreover, we also studied the change of optical properties of GaAs samples after high-energy electron irradiation.

Thus, this work studied the effect of high-energy electron irradiation on the structural and optical properties of a semi-insulating GaAs (001) wafer. The GaAs wafers were irradiated with different electron energies of 10, 15, and 20 MeV and with various irradiation doses of 0.0, 0.5, 1.0, and 2.0 MGy. Structural change in microscopic and macroscopic scales of GaAs wafers after electron irradiation was clarified by high-resolution X-ray diffraction (HRXRD) measurement in three scan modes of the 2θ-ω scan, ω-rocking scan, and reciprocal space mapping (RSM), focus ion beam scanning electron microscopy (FIB-SEM), atomic force microscopy (AFM), contact angle goniometer and Raman spectroscopy. The results showed that electron irradiation could induce surface and crystalline changes. Furthermore, the change of optical property of GaAs sample after electron irradiation was investigated using UV-VIS spectroscopy. Our results found that electron irradiation treatment can improve the optical property of GaAs sample until electron energy increases to 20 MeV and electron dose of 1.0 MGy. In addition, the effect of structural change in microscopic and macroscopic scales on the optical property of GaAs was systematically discussed in detail.

## 2. Materials and Methods

### 2.1. GaAs Wafer

The commercial GaAs wafer was purchased from Wafer Technology Ltd., United Kingdom. The semi-insulating 625-thick (001) GaAs wafer was cut into 1 × 1 cm^2^ and cleaned oxide layer with DI water and methanol several times. The cleaned GaAs samples were exposed to a high-energy electron beam in different potential energies and doses to investigate the GaAs wafer’s radiation hardness against high energetic electrons. The GaAs wafer has a smooth surface with a small RMS roughness (R_RMS_) of 5.993 nm and a hydrophobic surface with a contact angle of 91.34°, as shown in Figure 1.

### 2.2. Electron-Beam Irradiation Facility

The study was carried out for irradiation at the Gems Irradiation Center (GIC), Thailand Institute of Nuclear Technology (TINT). High-energetic electron delivery was performed using the Mevex electron accelerator (Mevex Corporation Ltd. MB 20-16, Stittsville, ON, Canada). The system consists of dual irradiation technology capabilities: X-ray and electron beam, which can produce high-energy electrons ranging from 8–25 MeV and accelerator power during 10–16 kW [6]. An aluminum collimator tailored the electron beam into a 1 × 1 cm^2^ square, which was suitable for each GaAs sample’s exact dimension. The GaAs samples were irradiated at room temperature under three different potential energies of 10, 15, and 20 MeV with irradiation dose rates ranging from 1.8–7.6 kGy/min to reach the accumulated doses of 0.5, 1.0, and 2.0 MGy, respectively.

### 2.3. Structural Features Analysis of GaAs Wafer

Changes in the crystal structure of the GaAs samples were studied with a high-resolution X-ray diffractometer equipped with a graded parabolic mirror combined with a 2-bounce monochromator of a channel cut Ge (220) crystal monochromator and a secondary Ge (220) crystal monochromator, which was positioned in front of the beam source and the detector, respectively. The HRXRD measurements were carried out using the Rigaku TTRAX III equipped with a CuK_α1_ beam source at a wavelength of 1.5406 Å. Three HRXRD scan modes: 2θ-ω scan, ω-rocking curve scan, and RSM around GaAs (004) plane, were acquired to determine lattice distortion, mosaic, and strain-relaxation of the samples. Lattice distortion was also verified by Raman spectroscopy measured at room temperature using a 633 nm He-Ne laser as the excitation light source in backscattering geometry with a power laser of 25 mW. Surface morphology of the irradiated samples was observed by the third generation of FEI’s Helios Nanolab Ultra High-resolution scanning electron microscope (SEM) equipped with focused ion beam (FIB) technology, FIB-SEM (FEI, Helios NanoLab G3 CX, Hillsboro, OR, USA) operating at an acceleration voltage of 15 kV. AFM was used with an XE-120 Park System AFM (Park System, Suwon, Korea) and non-contact silicon cantilever (PPP-NCHR, Nanosensors™, Neuchatel, Switzerland) with a spring constant of ~0.16 N m^−1^ and a resonant frequency of 5 kHz were used for imaging. A scanning speed was set to 0.7 Hz. The microscope was equipped with a piezo scanner with a maximum scan range of 2 × 2 μm^2^. Analysis of the AFM images was performed with Park Systems XEI image analysis software. A contact angle goniometer (First Ten Angstroms FTA1000 drop Shape Instrument B Frame Analyzer System, Portsmouth, VA, USA) was carried out to measure the contact angle between the dropped water and the sample surface to verify surface wettability.

### 2.4. Optical Properties Analysis of GaAs Wafer

Optical absorption measurements were carried out at room temperature using a UV-1800 spectrophotometer (UV-1800, Shimadzu, Tokyo, Japan) located at the Faculty of Science, Khon Kaen University. The recorded spectra were measured on the GaAs samples oriented perpendicular to the c-axis before and after irradiation in wavelengths ranging from 200 to 2500 nm by setting a spectral resolution of 0.2 nm. GaAs samples’ bandgap energy (*E*_g_) was evaluated from the Tauc plot in Appendix A. The *E*_g_ of the non-irradiated GaAs sample is approximately 1.373 eV at room temperature with low absorption intense UV-vis spectra of 18.59%.

## 3. Results and Discussion

The surface morphology of the irradiated GaAs samples is characterized by FIB-SEM and AFM, as shown in Figure 2 and Figure 3, respectively. Figure 2 compares the surface morphology of the irradiated samples at 10, 15, and 20 MeV electrons with irradiation doses of 0.5, 1.5, and 2.0 MGy, respectively. The 10 MeV irradiated GaAs samples (Figure 2a–c) have a smooth surface entire irradiation dose from 0.5 to 2.0 MGy compared to the non-irradiated GaAs surface (Figure 1a). The 15 MeV irradiated GaAs samples (Figure 2d–f) also have smooth surfaces at the lowest irradiation dose of 0.5 MGy. However, by increasing the irradiation dose to 1.0 and 2.0 MGy, the irradiated surfaces have some holes spreading on their surfaces. In the case of the 20 MeV irradiated GaAs samples (Figure 2g–i), their surface distinguishes composed of numerous holes spreading the entire surface with hole densities of 200, 72, and 32 nm^−2^, corresponding to an irradiation strength of 0.5, 1.0, and 2.0 MGy, respectively, with larger average hole size. The average hole size is increased from 0.015 (0.5 MGy) to 0.027 (1.0 MGy) and 0.066 nm (2.0 MGy) with a longer exposure time (higher irradiation dose). The holes appearing on the surface may be due to a melt ability of the electrons’ high energy, which caused some atoms (Ga or As) to escape from the lattice site. This occurrence of a hole on the irradiated surface was observed in our previous work, which was induced by gamma irradiation on the GaAsN surface [49].

Figure 3 shows AFM images of the irradiated GaAs samples with a 2 × 2 µm^2^ scan area. The surfaces of all irradiated GaAs samples (Figure 3) appear rougher surface with increased electron energy and electron dose, compared to the non-irradiated sample (Figure 1b), in agreement with the FIB-SEM result (Figure 2). This observation agrees well with Shen et al.’s work [37] and our previous work [49]. Their works discovered the irradiated samples had a rougher surface with increasing gamma irradiation dose. The variation of the R_RMS_ values with electron energy and electron dose is shown in Figure 4a. Based on the observation from the graph, the R_RMS_ values of the irradiated surface tend to increase with increasing electron energy and electron dose. The irradiated GaAs samples at an electron energy of 10 MeV have a smooth surface with a low R_RMS_ value of 8.385 (0.5 MGy), 8.615 (1.0 MGy), and 9.984 (2.0 MGy) nm, respectively, compared to the other samples. The 20 MeV irradiated GaAs sample obtained the largest R_RMS_ value of 14.944 nm at the highest irradiation dose of 2.0 MGy, corresponding to its largest hole size of 0.066 nm.

Moreover, the surface wettability of the irradiated samples was determined by the water droplet shapes on GaAs surfaces, as shown in Appendix A. The contact angles between the dropped water and the irradiated GaAs surface were plotted in Figure 4b. It is seen that the contact angle between the dropped water and the irradiation GaAs surface decreases with increasing irradiation energy and irradiation dose. The contact angle between the dropped water and the irradiated GaAs surface is in the range of 64.78°–87.17° less than that of the non-irradiated sample (91.34°). It implies that the irradiated samples have a higher hydrophilic surface than the non-irradiated sample. This may be because electron irradiation can cause an increase in the surface energy of the irradiated surface [50] due to its high surface roughness. Especially at the highest electron energy of 20 MeV and the highest irradiation dose of 2.0 MGy, the irradiated GaAs surface has the smallest contact angle of 64.78°, corresponding to the highest R_RMS_ value of 14.944 nm.

To verify the crystalline quality of the non-irradiated and irradiated GaAs samples, three scan modes of HRXRD measurements of symmetric (004) peak, including 2θ-ω scan, ω-scan rocking curve, and RSM modes, were performed. Figure 5a–c shows HRXRD 2θ-ω profiles of all samples. The diffraction peak of the GaAs (004) plane was adjusted to be constant at 66.049° for all samples. Especially at the highest electron energy of 20 MeV and the highest electron dose of 2.0 Mgy, there is a shoulder peak at a high diffracted angle. The full width at half maximum (FWHM) of the GaAs (004) peak from the 2θ-ω scan curve is shown in Figure 5d. At electron energies of 10 and 15 MeV, the FWHM of the GaAs (004) peak tends to decrease with increasing irradiation doses, indicating improving crystal uniformity due to thermal annealing from the high-energy electron beam. This may be due to interstitial atoms moving to a lattice, or vacancy sites, improving lattice spacing uniformity. At electron energy of 20 MeV, FWHM of the GaAs (004) peak tends to decrease at the low irradiation doses of 0.5 and 1.0 Mgy, resulting in improved crystallinity. However, at an irradiation dose of 2.0 Mgy, the FWHM of the GaAs (004) peak is larger than that of the other samples. This indicates that a high irradiation dose can increase the non-uniformity of (004) plane lattice spacing. This result suggests that a longer exposure time at a higher electron energy of 20 MeV can break Ga-As bonds, induce Ga or As vacancies, induce interstitial atoms and induce amorphous formation structure, causing increase lattice spacing non-uniformity.

Figure 6a–c shows the HRXRD ω-scan rocking curve around the GaAs (004) plane of all GaAs samples. FWHM of the (004) reflection peak measured from all GaAs samples is shown in Figure 6d. FWHM of (004) rocking curves for the irradiated samples is larger than that of the non-irradiated sample. It is known that an increase in FWHM of the (004) rocking curve indicates an increase in mosaic structures corresponding to distributions of crystallographic orientations. This mosaic structure may be caused by amorphous formation. This high FWHM means a low crystal quality of the irradiated GaAs samples. To further confirm the crystal quality of all GaAs samples, the RSM mode of the HRXRD scan continuous around the (004) plane in both 2θ-ω and ω-rocking modes was performed, and the result is shown in Figure 7. The 2θ-ω axis (*x*-axis) is related to the parallel interplanar spacing. From the maps, the calculated lattice parameter of the GaAs wafer is 5.653 Å, coincident with the lattice parameter of a free-standing GaAs (5.653 Å), indicating that the irradiated GaAs wafer is relaxed in all samples. This corresponds to the circular-like shape of the contour plot of the RSM non-irradiated sample (Figure 7a). It is seen that the peak width in the 2θ-ω axis tends to decrease with increasing irradiation energy indicating improved uniformity of lattice spacing. This result corresponds to a single line of 2θ-ω scan mode in Figure 5. In ω-rocking (*y*-axis), Δω of the irradiated samples (Figure 7b–d) is larger with increasing irradiation energy, indicating an increase in mosaic structure in agreement with a single line of ω-rocking curve mode in Figure 6. This mosaic may be due to Ga or As atoms escaping from the lattice site and amorphous formation structure induced by thermal heating from high-energy electron bombardment. These phenomena affect microscopic scales inside the crystal structure, as evidenced by the HRXRD result and the macroscopic scales observed by surface roughness from FIB-SEM and AFM images.

Figure 8 shows Raman spectra of the GaAs samples irradiated by electron energies of (a) 10, (b) 15, and (c) 20 MeV. Raman shifts at a wavenumber of 267.3 and 293.0 cm^−1^ corresponding to transverse-optical (TO) and longitudinal-optical (LO) phonon modes of GaAs [51] were observed for the non-irradiated sample. Based on the Raman selection rules, TO modes are not allowed for zincblende structures in backscattering geometry. This deviation from the selection rules is due to a breakdown in the long-range order of the zincblende crystal related to lattice distortion. However, the TO mode of all GaAs samples can be observed, and its intensity and peak width increase with increasing irradiation dose, indicating higher lattice distortion (Figure 8a) induced by loss of Ga or As atom after irradiation. For electron energies of 15 and 20 MeV, as shown in Figure 8b,c, it can be clearly seen that the Raman intensity obtained from the TO mode is stronger than that obtained from the LO mode with increasing irradiation dose. Moreover, a new peak appears at a wavenumber of around 200 cm^−1^ due to disorder-activated longitudinal acoustic (DALA) [52], as can be observed for high irradiation doses of 1.0 to 2.0 Mgy (electron energy of 15 MeV) and 0.5 to 2.0 Mgy (electron energy of 20 MeV). These results strongly imply that the irradiated GaAs samples at higher electron energies (15 and 20 MeV) have a larger lattice distortion than the irradiated sample at low irradiation energy (10 MeV) induced by a larger amount of Ga or As vacancies, interstitial atoms, amorphous formation, and mosaic structure domain. These results coincide well with the HRXRD curve in three scan modes, as evidenced in Figure 5, Figure 6 and Figure 7.

Figure 9 shows UV-VIS spectra of the GaAs samples irradiated with electron energies of (a) 10, (b) 15, and (c) 20 MeV. The *E*_g_ of the irradiated GaAs samples, evaluated by Tauc plot, as shown in Appendix A, was 1.364 to 1397 eV. As seen in Figure 9a, for electron energy of 10 MeV, the absorption band edge of the irradiated samples is sharper than that of the non-irradiated sample for all irradiation doses without a band tail. This may be due to the high crystal uniformity of the irradiated sample after electron irradiation treatment induced a loss of band tail of UV-Vis spectra. In addition, the absorption spectra of the irradiated samples have a higher intensity than that of the non-irradiated sample. This indicates that the irradiated samples can collect more light intensity than the non-irradiated sample. Significantly, the irradiated GaAs sample at 10 MeV electron for 2.0 MGy dose can absorb UV-Vis spectra of 122.66%, about 7 times the non-irradiated sample. Higher absorption spectra of the irradiated samples may be due to increased lattice uniformity caused by reducing lattice defects (Ga or As vacancies, interstitial atoms) in the crystal structure agreeing with HRXRD and Raman scattering results or less reflected light on rough surfaces. The rough surface with a hole of the irradiated samples may act as nano-grating for protection against reflected light, as observed in Das et al.’s work [53,54]. In the case of electron energies of 15 and 20 MeV in Figure 9b,c, the irradiated samples’ absorbance intensity also tends to be higher than that of the non-irradiated sample and increases with increasing irradiation dose. These high absorption spectra of the irradiated samples may help create large amounts of electron-hole pairs and carrier mobility in the irradiated sample. These could be improved the conversion efficiency of GaAs-based solar cells in space applications or environments of high-energy particle bombardment. However, comparing the absorption spectra of the irradiated sample at the 2.0 Mgy irradiation dose, the absorption spectra respective decrease with increasing irradiation energies from 10 to 20 MeV. This may be due to the higher crystal non-uniformity of the 20 MeV irradiated GaAs sample than that of the 10 and 15 MeV irradiated samples agreeing with HRXRD and Raman scattering results. However, the absorption spectra of the 20 MeV irradiated sample (Figure 9c) increase with the irradiation dose. The highest absorption spectra can still be obtained from the 2.0 MeV irradiation dose sample, while the crystal structure has the highest non-uniformity. This may be because its largest roughening surface (R_RMS_ = 14.944 nm) can reduce the reflection of light on the irradiated surface, increasing absorption spectra. Our results show that the irradiated GaAs samples still have high crystal quality and optical properties until irradiation energy and irradiation dose are as high as 20 MeV and 1.0 Mgy, respectively. Our results indicate that the optical property (absorption spectra) of the irradiated sample depends on its structural properties in microscopic (crystal uniformity involving lattice defects such as vacancies and interstitial) and macroscopic (surface roughness) properties.

## 4. Conclusions

This work systematically studied the effect of electron irradiation on GaAs’ structural and optical properties under different electron energies of 10, 15, and 20 MeV and electron doses ranging from 0–2.0 MGy. The surface morphology of the irradiated GaAs samples showed roughness surface, with many holes spreading on their surface. The irradiated GaAs sample at an electron energy of 20 MeV and electron dose of 2.0 MGy obtained the largest surface roughness of 14.944 nm and the largest hole size of 0.066 nm. The surfaces of the irradiated GaAs samples became more hydrophilic compared with the non-irradiated surface, with the lowest contact angle of 64.78° observed in the sample treated with irradiation energy of 20 MeV and irradiation dose of 2.0 MGy. HRXRD and Raman scattering results showed the crystallinity of the irradiated GaAs samples could be improved by low electron energies of 10 and 15 MeV. However, the crystallinity of the irradiated GaAs sample at the highest electron energy of 20 MeV tended to be damaged lattice uniformity with increasing irradiation strength to 2.0 MGy. After irradiation treatment, the bandgap energy of GaAs can be tuned, ranging from 1.364 to 1.397 eV. Moreover, the absorption spectra of the irradiated sample were higher than that of the non-irradiated sample due to improving crystal uniformity and its roughening surface. This result revealed that electron irradiation treatment could improve the structural and optical properties of the irradiated GaAs samples when electron energy and electron dose did not exceed 20 MeV and 1.0 MGy. Finally, outcome results can visualize the effect of high-energy electron irradiation on microscopic defects in the crystal structure of GaAs and further clarify the changes in the macroscopic properties of this material to utilize it for a particular application.

## Figures and Tables

**Figure 1 materials-15-05897-f001:**
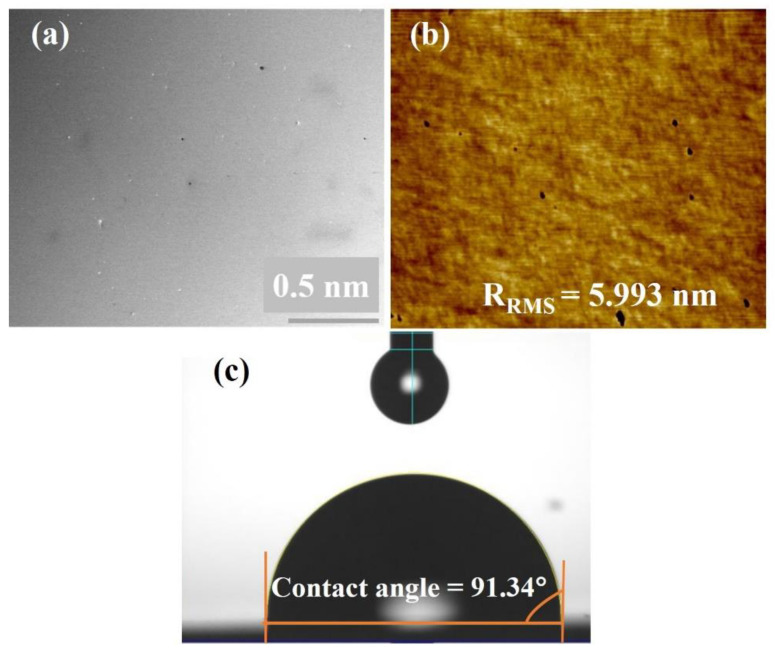
(**a**) FIB-SEM image; (**b**) AFM image; and (**c**) water dropped on the non-irradiated GaAs sample.

**Figure 2 materials-15-05897-f002:**
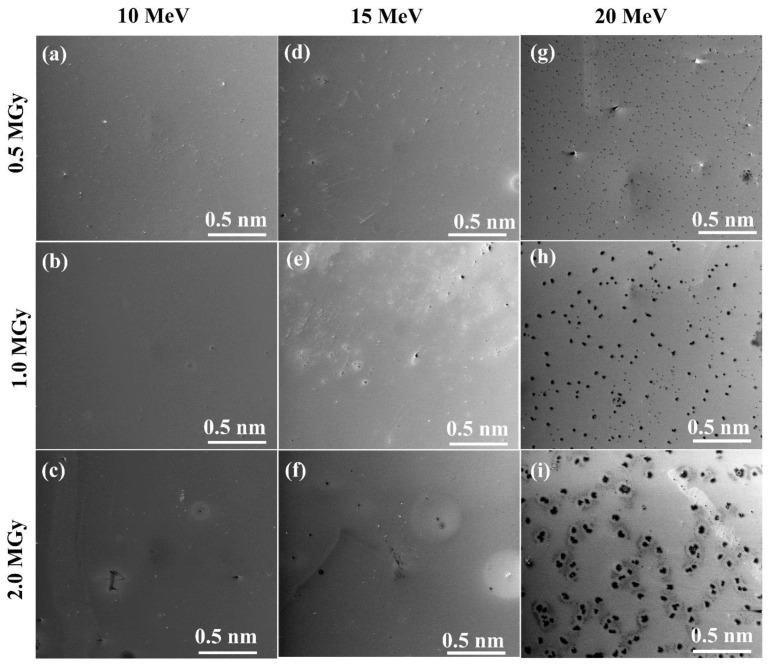
FIB-SEM images of the irradiated GaAs samples with electron energies of 10 (**a**–**c**), 15 (**d**–**f**), 20 (**g**–**i**) MeV for electron doses of 0.5, 1.0, and 2.0 MGy, respectively.

**Figure 3 materials-15-05897-f003:**
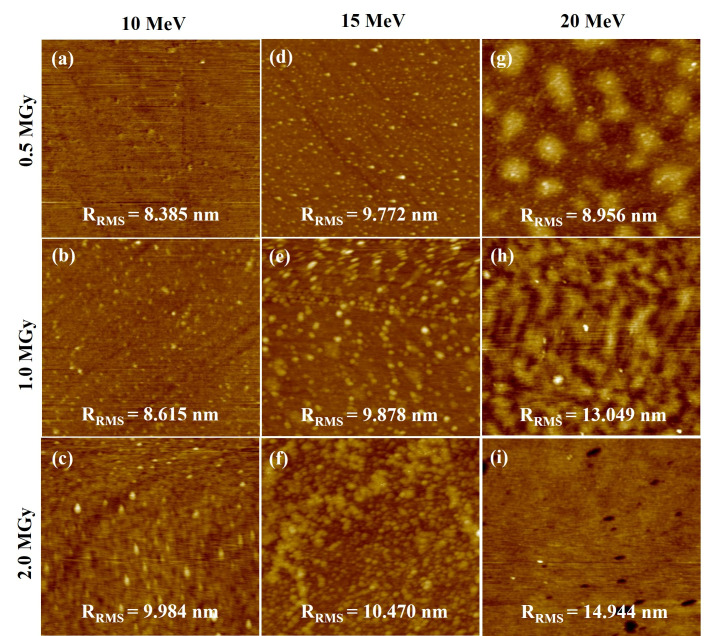
AFM images of the irradiated GaAs samples with electron energies of 10 (**a**–**c**), 15 (**d**–**f**), 20 (**g**–**i**) MeV for electron doses of 0.5, 1.0, and 2.0 MGy, respectively.

**Figure 4 materials-15-05897-f004:**
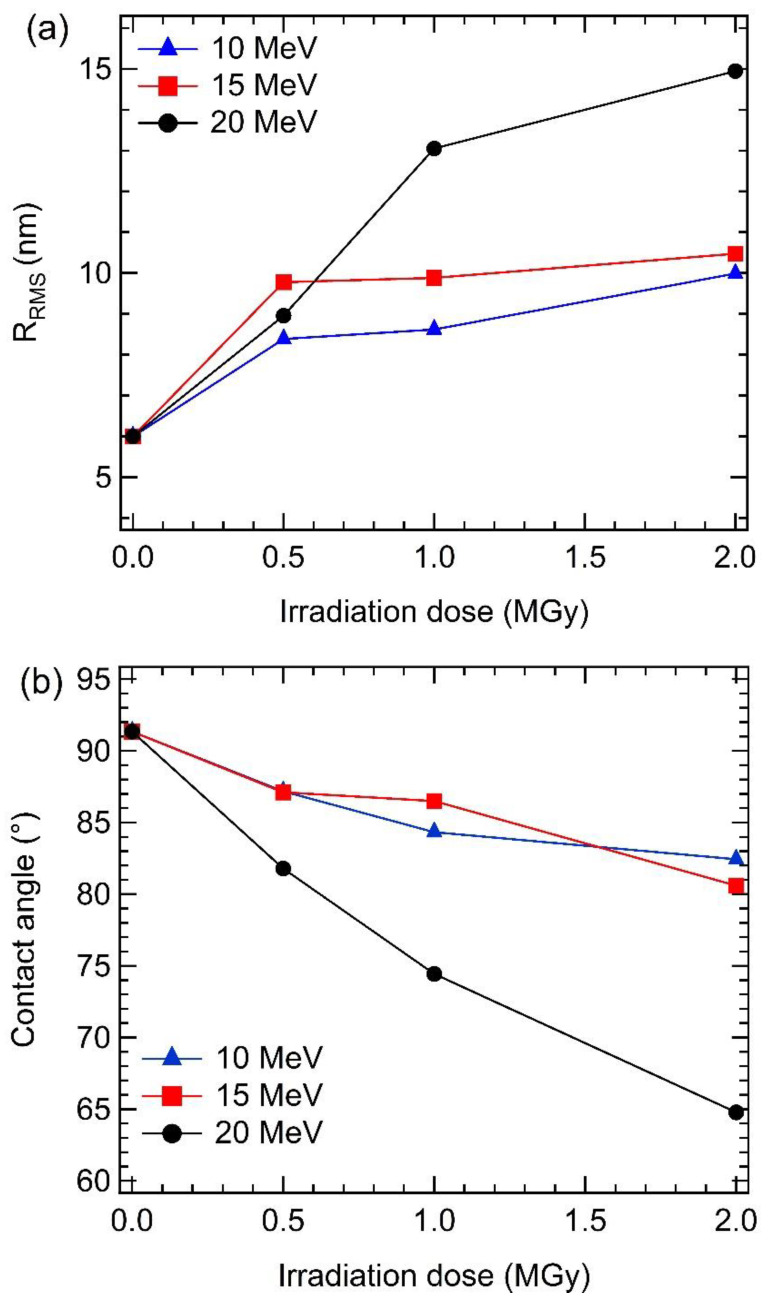
(**a**) R_RMS_ values and (**b**) contact angle of the irradiated GaAs samples with electron energies of 10, 15, and 20 MeV at electron doses of 0.5, 1.0, and 2.0 MGy.

**Figure 5 materials-15-05897-f005:**
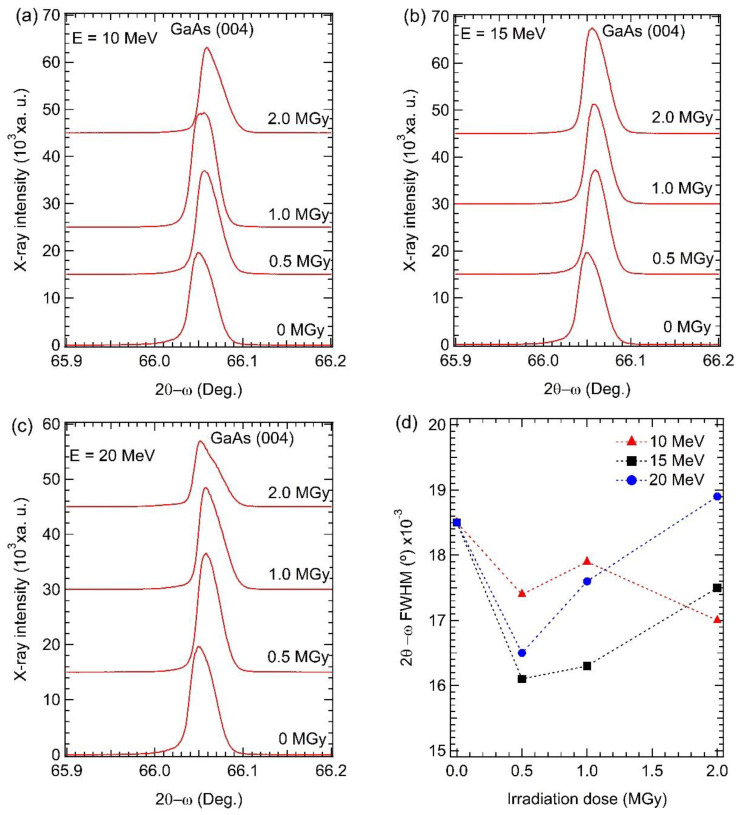
Symmetric (004) 2θ-ω curves of the non-irradiated and the irradiated GaAs with different irradiation strengths at an electron energy of (**a**) 10; (**b**) 15, and (**c**) 20 MeV; (**d**) The full width at half maximum (FWHM) of GaAs (004) peak.

**Figure 6 materials-15-05897-f006:**
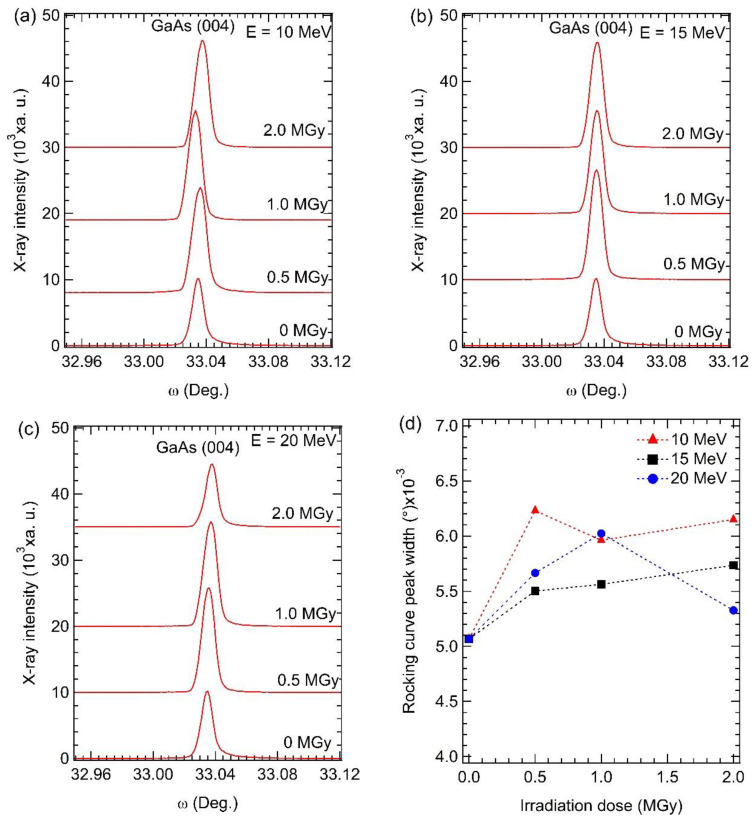
(004) HRXRD rocking curves of the non-irradiated and irradiated GaAs with different irradiation strengths at an electron energy of (**a**) 10; (**b**) 15; and (**c**) 20 MeV; (**d**) FWHM of (004) rocking curves.

**Figure 7 materials-15-05897-f007:**
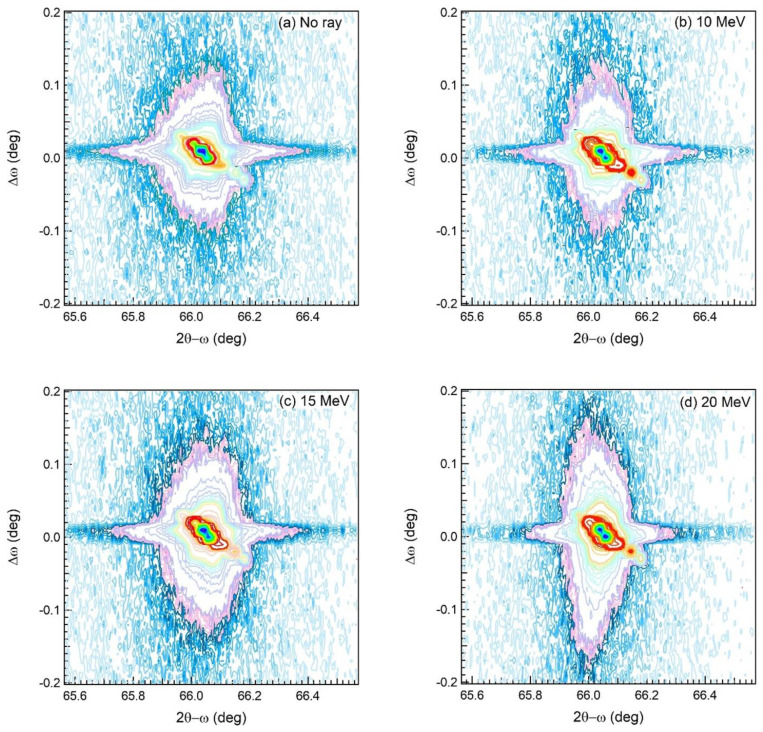
Symmetric (004) RSM of the non-irradiated GaAs (**a**), the irradiated GaAs at the highest irradiation dose of 2.0 MGy at an electron energy of (**b**) 10; (**c**) 15; and (**d**) 20 MeV.

**Figure 8 materials-15-05897-f008:**
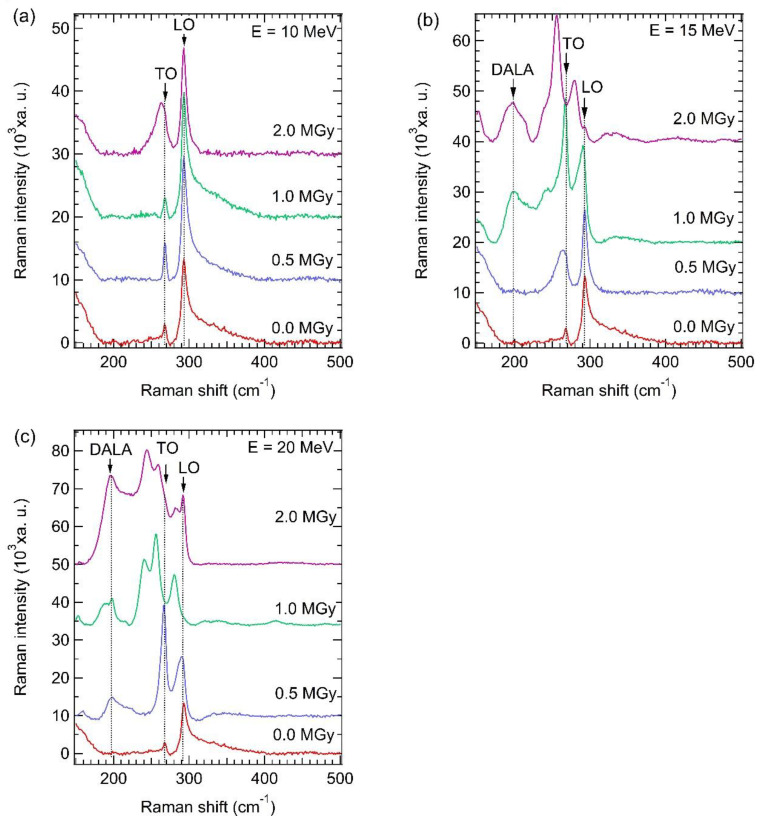
Raman spectra of GaAs samples irradiated with electron beam at (**a**) 10; (**b**) 15; and (**c**) 20 MeV.

**Figure 9 materials-15-05897-f009:**
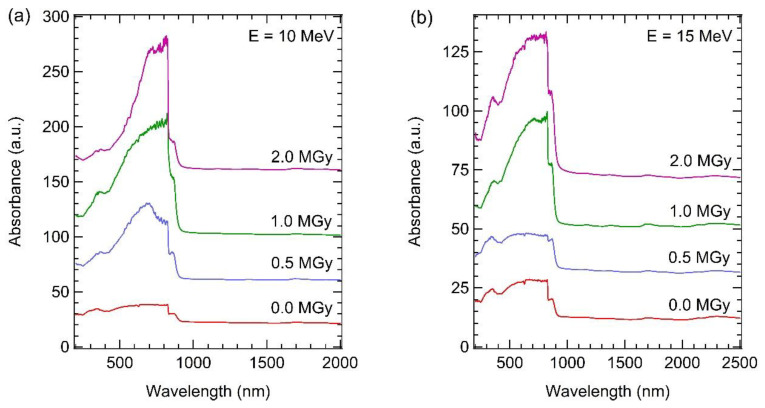
UV-VIS spectra of GaAs samples irradiated with different energy of (**a**) 10; (**b**) 15; and (**c**) 20 MeV.

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
