# Peer review of "Investigation of Radiation Effect on Structural and Optical Properties of GaAs under High-Energy Electron Irradiation"

_materials, 2022, doi:10.3390/ma15175897_

Round 1

Reviewer 1 Report

The work reported deals with the systematic investigation of the changes in structural and optical properties of semi-insulating GaAs (001) wafer under high-energy electron. GaAs wafers were exposed to high-energy electron beams under different energies of 10, 15, and 20 MeV for absorbed doses ranging from 0 – 2.0 MGy. The study observed the changes in bandgap energy of GaAs samples after irradiation with 10-MeV, 15-MeV, and 20-MeV electrons. The bandgap energy was found in the 1.364 – 1.397 eV range, and the observed intense UV-VIS spectra were higher than in non-irradiated samples. Overall, the work is interesting and can be considered for publication after addressing minor comments.

1.         I recommend the authors should include some new references from Materials Journal(MDPI) which are relevant to use in the introduction as well as results and discussion.

2.         Reformat the figures to match the standards of the Materials journal.

3.         Check the manuscript for grammatical errors and correct them.

4.         Emphasize the experimental results in the conclusion part of the manuscript.

Reviewer 2 Report

The manuscript entitled " Investigation of radiation effect on structural and optical properties of GaAs under high-energy electron irradiation". The experimental work is interesting. The crystal structure, surface morphology, and optical properties of as-prepared samples have been studied. In my opinion, this work is interesting and has a certain reference in the development for the application in high-speed semiconductor devices, high- power microwave and millimeter-wave devices, optoelectronic devices, medical detectors, and imaging devices. However, there are some remarks that should be taken into consideration by the authors in order to raise this article to a good level for publication in Materials.

The suggested modifications are listed as follows:

1. The first occurrence of an acronym should be explained in detail. Such as: RMS.

2. “10-MeV, 15-MeV, and 20-MeV” should be revised as “10, 15, and 20 MeV”.

3. In the introduction, the study of GaAs affected by electron irradiation should be supplemented. A considerable amount of literature has studied the effect of electron irradiation on the optical properties of GaAs. Careful summarization is needed to extract the originality of this manuscript.

4. The ordinate values in Figure S1 should be given. Eg value is the focus value of the slope of the curve and the abscissa (when the ordinate is 0).

5. You can try to explain the mechanism of irradiation. Some scholars believe that irradiation of semiconductor materials with high-energy ions will lead to its amorphous formation.
